# Coherent exciton-vibrational dynamics and energy transfer in conjugated organics

Tammie R. Nelson[1], Dianelys Ondarse-Alvarez[2], Nicolas Oldani[2], Beatriz Rodriguez-Hernandez[2], Laura Alfonso-Hernandez[2], Johan F. Galindo [3], Valeria D. Kleiman[4], Sebastian Fernandez-Alberti[2], Adrian E. Roitberg[4] & Sergei Tretiak [1]

Coherence, signifying concurrent electron-vibrational dynamics in complex natural and man-made systems, is currently a subject of intense study. Understanding this phenomenon is important when designing carrier transport in optoelectronic materials. Here, excited state dynamics simulations reveal a ubiquitous pattern in the evolution of photoexcitations for a broad range of molecular systems. Symmetries of the wavefunctions define a specific form of the non-adiabatic coupling that drives quantum transitions between excited states, leading to a collective asymmetric vibrational excitation coupled to the electronic system. This promotes periodic oscillatory evolution of the wavefunctions, preserving specific phase and amplitude relations across the ensemble of trajectories. The simple model proposed here explains the appearance of coherent exciton-vibrational dynamics due to non-adiabatic transitions, which is universal across multiple molecular systems. The observed relationships between electronic wavefunctions and the resulting functionalities allows us to understand, and potentially manipulate, excited state dynamics and energy transfer in molecular materials.

[1] Theoretical Division, Center for Nonlinear studies and Center for Integrated Nanotechnologies, Los Alamos National Laboratory, Los Alamos, NM 81545, USA. [2] Universidad Nacional de Quilmes/CONICET, Roque Saenz Peña 352, B1876BXD Bernal, Argentina. [3] Department of Chemistry, Universidad Nacional de Colombia, Av. Cra 30 # 45-03, Bogotá 111321, Colombia. [4] Department of Chemistry, University of Florida, Gainesville, FL 32611, USA. Correspondence and requests for materials should be addressed to S.F-A. (email: sfalberti@gmail.com) or to S.T. (email: serg@lanl.gov)

Coherence is defined as an in-phase evolution of specific degrees of freedom. In electronic dynamics of materials controlled by quantum mechanical laws, coherence frequently appears as amplitude correlations in delocalized wavefunctions and manifests itself in interference patterns persisting over long time-scales[1]. Formally, quantum-mechanical coherences are defined as off-diagonal elements in the density matrix and, as such, they are not directly observable but can be derived from the presence of measurable spectroscopic signals. About a decade ago, persistent quantum coherence was discovered in the initial stage of photosynthesis across several highly structured biological light-harvesting complexes[1–7]. Later, similar phenomena were observed across many other molecular and nanostructured materials[8–12]. While the initial reports had attributed the observed dynamics to unexpectedly long-lasting electronic coherences, later investigations linked it to the interplay between both electronic and vibrational degrees of freedom[13,14] and it was broadly hypothesized that oscillatory evolution of delocalized electronic wavefunctions can improve transport of energy and charge carriers for light-harvesting, lighting, and other optoelectronic applications[1,3,8,15,16]. The change in thinking towards more complex interaction between vibrations and electronic coherences was particularly prevalent in the realm of photobiology[17], where commonly employed models treat vibrations as quantum degrees of freedom[18–23].

Most of the systems studied above belong to the "intermediate coupling regime", when the electronic and vibrational couplings are comparable[9]. The transport processes following photoexcitation are concomitant to non-radiative relaxation, when the system dissipates the excess of electronic energy into heat. During this internal conversion, energy typically flows from the electronic to vibrational degrees of freedom via two distinct mechanisms. When electronic states are well separated, the system can relax adiabatically downhill on a single potential energy surface within the Born–Oppenheimer framework. Alternatively, when electronic states are close in energy, the Born–Oppenheimer approximation breaks down and non-adiabatic evolution takes place when the electronic state (and the respective potential energy surface) changes during the dynamics[8,14]. This is a common scenario for energy transfer. Here, one extreme includes strong electronic couplings leading to fully delocalized states and an efficient band-like transport (such as the case of classic semiconductors)[24]. Another extreme includes highly disordered materials with large vibrational coupling that limits transport to the incoherent hopping-like random walk regime[24]. Interestingly, for materials in the "intermediate coupling regime", there exists an ample amount of spectroscopic evidence for robust coherent electron-vibrational dynamics, which persists over long (frequently picosecond) timescales at ambient conditions, in spite of structural disorder, noise, and environmental fluctuations that may be present. Subsequently, several recent reviews[1–3,7] suggested that coherence is a highly non-trivial and very important factor, which can be used to achieve specific functionalities in chemical and biological systems provided that underpinning design principles[25,26] are well understood.

Here, we show how coherent exciton-vibrational dynamics emerges in photoactive molecular systems due to non-adiabatic (non-Born–Oppenheimer) transitions between excited states. Previous studies recognized the importance of symmetry of vibronic coupling between different electronic states in resonant transitions[27,28], and electron[29,30] and energy[31,32] transfer rates. Here, we are exploring its effect on coherent electron-vibrational dynamics. This phenomenon is ubiquitous as it follows from simple interplays between localizations and symmetries of the wavefunctions. Namely, non-adiabatic transitions between excited states induce the spatial coherence between the eigenstates of

the electronic molecular Hamiltonian, which are dynamically modulated by classical vibrational motions. Since such transitions are often not a singular event and can persist for some time, observed dynamics is strongly dependent on the system in question. We first present a simple conceptual model rationalizing the asymmetric form of the derivative non-adiabatic coupling (NAC) vector responsible for driving transitions between excited states. This, in turn, initiates a specific vibrational excitation modulating the wave-like localized–delocalized motion of the electronic wavefunction. We further demonstrate universality of these phenomena by inspecting photo-induced dynamics in several common cases for organic conjugated materials. These include a linear oligomer, nano-hoop, tree-like dendrimer, and molecular dimer. In all these molecules, ultrafast dynamics and exciton transport is directly simulated using our atomistic non-adiabatic excited-state molecular dynamics (NEXMD) package[33]. Coherent dynamics observed in these systems persists on the timescale of hundreds of femtoseconds at room temperature and in the presence of a bath, which agrees with experimental spectroscopic reports on various materials. Here, coherences are controlled by electronic and vibrational coupling unique to the chemical composition and structural conformation. Such general behavior suggests common strategies for manipulating electronic functionalities, such as charge and energy transport, in both natural and synthetic systems.

## Results

**Alternating wavefunction symmetry**. To establish a conceptual framework, we recall that photo-induced electronic processes in realistic molecular systems predominantly involve a broad manifold of excited states. Subsequently, avoided and unavoided (e.g., conical intersections) crossings between potential energy surfaces (PESs) define the dynamics, where non-adiabatic transitions between states (internal conversion) are commonly occurring due to a breakdown of the Born–Oppenheimer approximation. Fig. 1a schematically shows two PESs with electronic wavefunctions labeled as $\Psi_1$ and $\Psi_2$ parametrically depending on multidimensional vibrational degrees of freedom **R**, where the colored box denotes the non-adiabatic coupling region. Excited state wavefunctions in low-dimensional organic materials such as conjugated polymers, branching structures, and molecular aggregates are excitons (electron-hole pairs interacting via Coulombic potential) with a large binding energy[16]. Importantly, the envelopes of these wavefunctions always adopt a standing wave pattern on the finite structures[34] following the particle (exciton) in a box model as shown in Fig. 1b. Here, the respective PES of the state defines the multi-dimensional potential landscape for a bound excitonic state. We further notice the presence of an alternating symmetry between wavefunction phases for sequential states in the band. While specific symmetry labels depend on the molecular geometry, here we will loosely use "symmetric" and "antisymmetric" labels as depicted in Fig. 1b. For example, the excited states in a prototype conjugated polymer polyacetylene have $A_g$ and $B_u$ symmetries (Fig. 1c), whereas $\Psi_1$ and $\Psi_2$ states in the molecular homo-dimer are symmetric and antisymmetric combinations of the parent $\phi_L$ and $\phi_R$ (left and right) monomer states ($\Psi_{1,2} = \phi_L \pm \phi_R$) within the Frenkel exciton model[35], thus illustrating the basis for our notations.

**Vibrational excitation initiated by internal conversion**. In a typical scenario for internal conversion (Fig. 1a), a photoexcited wavepacket goes through the crossing region to transition from the upper to the lower PES. Such processes are usually described via semiclassical models establishing consistent propagation of quantum (electrons) and classical (nuclei) degrees of freedom in

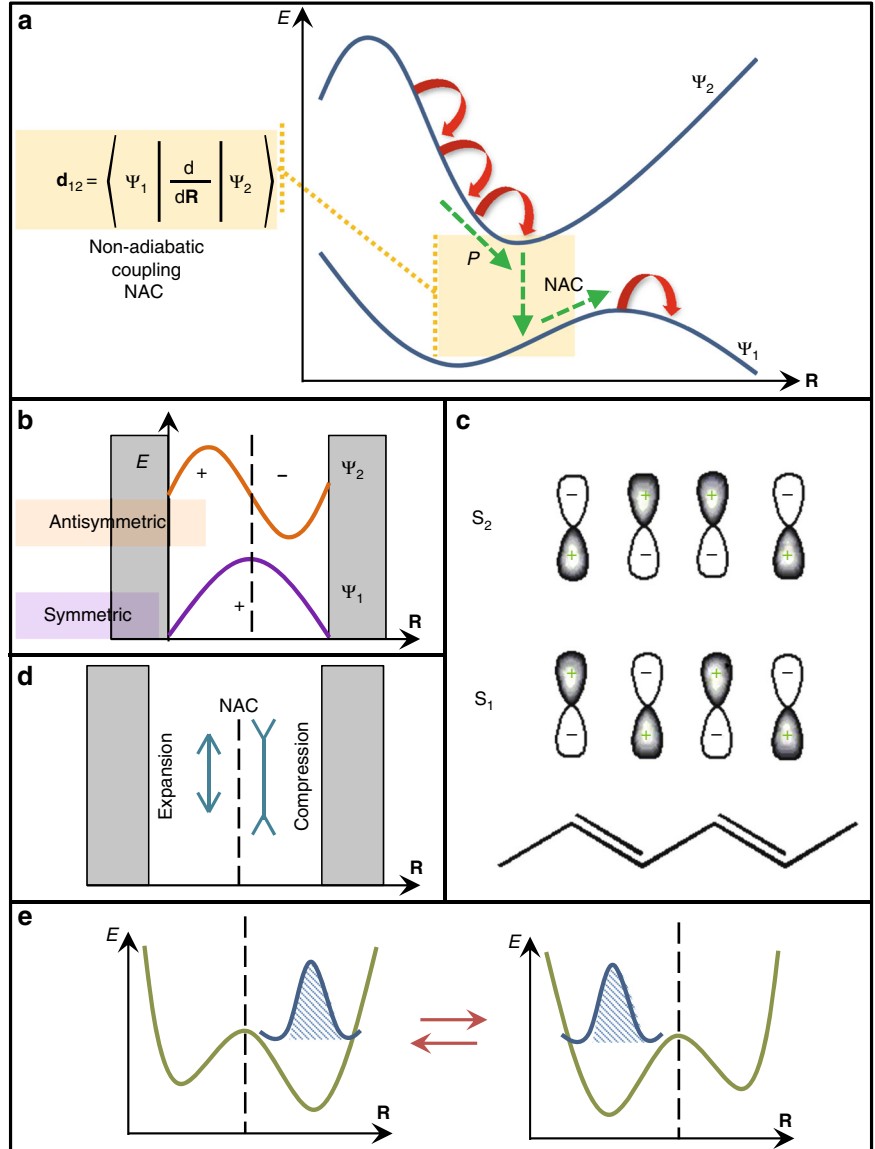

**Fig. 1** Electronic and vibrational coupling during internal conversion. **a** Schematic representation of dynamics through the non-adiabatic region (colored box) on two potential energy surfaces defined by electronic wavefunctions $\Psi_1$ and $\Psi_2$ with dependence on electronic degrees of freedom **R**. During internal conversion, the efficiency of the transition of the photoexcited wavepacket from the upper to the lower surface is driven by the derivative non-adiabatic coupling $\mathbf{d}_{12}$. On the upper surface, the wavepacket is pushed towards the crossing by the Pechukas force, $P$, acting in the direction of the non-adiabatic coupling vector (NAC). **b** The wavefunction on a finite molecule adopts a standing wave pattern according to the particle (exciton) in a box model exhibiting either "symmetric" ($\Psi_1$) or "antisymmetric" ($\Psi_2$) form. The non-adiabatic transition from $\Psi_2$ to $\Psi_1$ corresponds to an antisymmetric-to-symmetric transition between neighboring wavefunctions. **c** The two lowest energy excited states in the polyacetelene conjugated polymer exhibit $A_g$ and $B_u$ symmetries. **d** The resulting vibrational excitation has an asymmetric form where the left and right part of the system experience structural deformations with opposite phase (expansion and compression). **e** Sloshing of the localized wavefunction between left and right sides of the double well potential is initiated by the asymmetric vibrational excitation which causes periodic modulations in the potential energy surface on the lower state

the non-adiabatic regime[33]. Erhenfest and surface hopping[36] are examples of such methods allowing explicit treatment of large molecular systems for which fully quantum dynamics is prohibitively expensive[8,37–39]. Alternative perturbative approaches[40–42] usually treat nuclei as an effective bath, and the self-energy due to coupling of the nuclei and electrons is usually defined in frequency space and is estimated by averaging over the nuclear motion, thus losing the explicit correlation. Such approaches have been extensively applied, for example, to biological light-harvesting systems[43,44]. In this study, instead, the correlation between the electronic and nuclear dynamics is explicitly included in real time, though non-adiabatic, coupling. Notably, across all methodologies, the derivative coupling NAC $\mathbf{d}_{12}$ (Fig. 1a) drives

the efficiency of the transition. First, the wavepacket on the upper surface in the non-adiabatic region experiences the so-called Pechukas force (**P**, Fig. 1a) in the direction of the NAC vector pushing the system towards the crossing[45]. Furthermore, upon non-adiabatic transition, the excess electronic energy is dispersed into the nuclear velocities in the direction of the NAC vector to enforce energy conservation. The direction of the NAC vector is highly significant and it represents the direction of the driving force acting along a unique normal mode direction throughout regions of strong coupling[46,47]. The fact that the direction of the NAC vector defines the flux of energy toward specific vibrations has been emphasized by Bittner et al.[48]. This provides a simple physical rationale for adjusting nuclear velocities along the

direction of the non-adiabatic coupling vector. These electronic-to-vibrational energy conversion principles were proven at various levels of theory[45,49]. Subsequently, the NAC vector defines a displacement for a specific vibrational state within a lower PES absorbing the excess electronic energy from transitions between excited states. A rigorous iterative search of this vibrational coordinate was recently reported for the state-to-state transitions in the case of electronic transfer[48]. In our conceptual example of an asymmetric-to-symmetric transition between neighboring wavefunctions (Fig. 1b), the NAC vector defined in Fig. 1a (and the resulting vibrational excitation) has a strictly asymmetric form. Namely, the left (right) part of the system undergoes expanding (contracting) structural deformation with opposite displacement (or phase) as shown in Fig. 1d. We expect that such vibrational excitation is related to the structural motions usually considered to be coupled to the electronic degrees of freedom such as C–C stretches and torsional librations. However, it is not directly associated with any of the vibrational normal modes of either the ground or any excited state, rather being a complex superposition of several normal modes, as was demonstrated in the case of charge transfer[48]. In the present examples, the non-adiabatic coupling vector is commonly spread among a small subset of normal modes (~2–5) such as C–C stretches and torsions. A typical spectral width within each subclass of modes is less than 0.05 eV. These modes become active experiencing a substantial increase in their vibrational energy during the process[50].

Finally, initiated by electronic relaxation, asymmetric vibrational excitation periodically modulates the electronic wavefunction motions on the lower PES. This leads to the "sloshing" of the localized wavefunction between "left" and "right" sides (see Fig. 1e) with possible intermittent spatial delocalizations across the double well potential. Thus, symmetries of the initial wavefunctions define the form of vibrational excitation emerging

after electronic relaxation, which, in turn, controls wave-like localization–delocalization motion of the final wavefunction underpinning synchronous vibronic dynamics in the excited state. The dynamics of long-lived ground state wavepackets in photosynthetic light-harvesting antennas has already been reported in experiment[19].

**Applications to molecular systems.** To validate this scenario in realistic materials, we further study four systems: a linear oligomer (Fig. 2a) representing conjugated polymer family[39], a nanohoop (Fig. 2b) prototyping circular geometry of ubiquitous photosynthetic complexes[38], a dendrimer (Fig. 2c) exemplifying branched artificial light-harvesting systems[37], and a dimer (Fig. 2d) signifying molecular crystals and aggregates[51]. We use our NEXMD package to simulate internal conversion following photoexcitation in all the systems at ambient conditions in the presence of a bath, as outlined in Methods.

While our calculations may involve higher lying excited states to mimic time-resolved spectroscopic probes, here we focus our analysis on the transition between the two lowest excited electronic states $S_2$ and $S_1$ ($S_3$ and $S_2$ states in the dendrimer). Fig. 2 displays the orbital plots of the transition densities (see Methods) taken at the ground state equilibrium geometry, which reflect spatial distributions of the excited state wavefunctions. We immediately recognize the "asymmetric–symmetric" motif (Fig. 1b) for $\Psi_1$ and $\Psi_2$ in all systems. In the dimer example, orbitals for one monomer are in-phase, whereas they are out-of-phase for the other, reflecting "+" and "−" wavefunction combinations as discussed above. As expected, NAC $\mathbf{d}_{12}$ vectors have the corresponding spatially asymmetric forms (Fig. 2a–d, bottom plots), conveying the vibrational excitation dynamically emerging due to electronic transition, in line with the schematic in Fig. 1d. Interestingly, the asymmetric form of NAC persists across all dynamical simulations as illustrated for the case of a

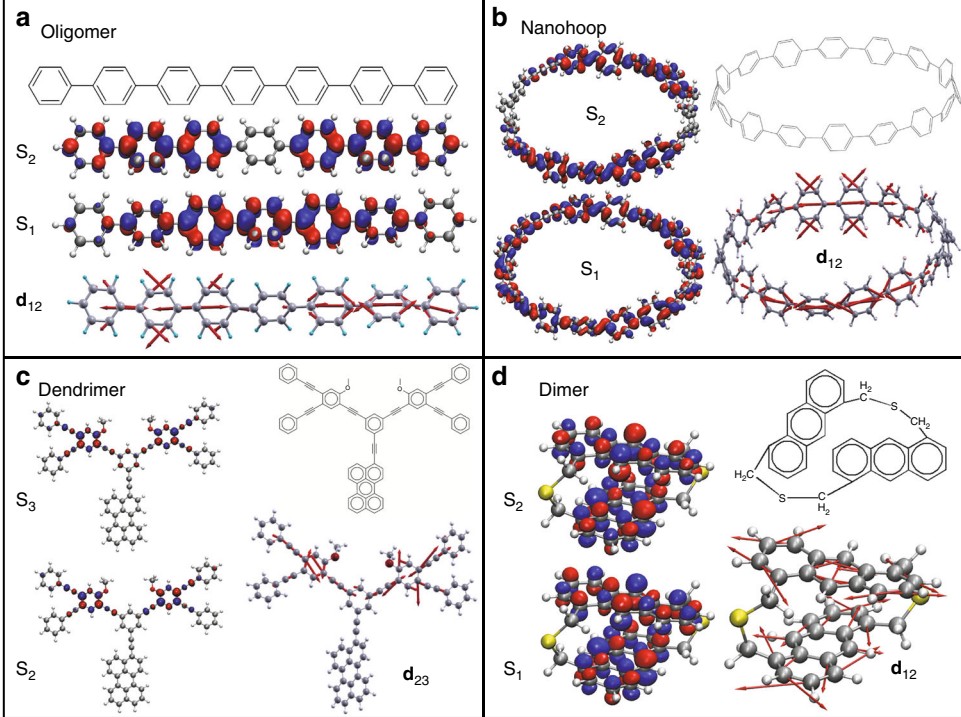

**Fig. 2** Characterization of four model systems. **a** polyphenylene oligomer, **b** cycloparaphenylene nanohoop, **c** branched phenylene ethynylene dendrimer, and **d** anthracene dimer (dithia-anthracenophane); Each panel contains the chemical structure, orbital plots of the transition densities for the two lowest excited states $S_2$ and $S_1$ ($S_3$ and $S_2$ in dendrimer) at the ground state equilibrium geometry, and the corresponding non-adiabatic coupling vector $\mathbf{d}_{12}$ ($\mathbf{d}_{23}$ in dendrimer)

dimer in Supplementary Fig. 1. It is clear that even for complex systems, the behavior described using our simple symmetry arguments holds true, as long as the system is composed of similar elementary building blocks (e.g., monomeric units in the case of a molecular aggregate or a crystal).

**Revealing spatially localized electronic states**. Having confirmed the foundations of our model, we further turn to the analysis of dynamical variables. Surface hopping algorithms underpinning the NEXMD simulations produce a trajectory ensemble. The meaningful observables, such as relaxation timescales, are then calculated as statistical averages. When non-radiative relaxation pathways lead the system to the regions where electronic states are not well separated and the Born–Oppenheimer (adiabatic) description is insufficient, values of the energy gaps lower than ~0.1 eV are expected. Plots of the relevant energy gap distributions for all systems (Supplementary Fig. 2) confirm that near the non-adiabatic transitions, the energy gap is small with a narrow distribution across the ensemble. Consequently, no superficial distinction between adiabatic and non-adiabatic regimes is made in our simulations, and the NEXMD trajectories are run for the entire duration of the dynamics starting from a ground state configuration instantaneously promoted to an excited state (Supplementary Fig. 3). For some systems, the wavepacket passes through the non-adiabatic region fast (e.g., oligomer) and the subsequent dynamics is essentially adiabatic. However, others (such as dendrimer or dimer) represent the case when the energy separation between excited states is comparable to the frequency of intramolecular motions and non-adiabatic dynamics persist for longer timescales. Consistent with the model in Fig. 1, the excitation is made to the low energy states that are less strongly overlapping compared to the dense energy manifold at higher energies that can be confirmed in the equilibrated absorption spectra in Supplementary Fig. 3 showing the density of excited states along with the excitation energy for each system. The electronic character of the initial excitation is mostly uniform across the ensemble. We start by inspecting a typical individual trajectory from the ensemble for each system. Fig. 3 shows transition density snapshots taken at different times during dynamics. In the case of the linear oligomer, the initial state is typically an asymmetric wavefunction with a single node along the backbone (Fig. 3a) directly conforming to Fig. 1b. Non-adiabatic transition leads to the sloshing of the wavefunction, following Fig. 1d, between left and right parts of the molecule (see snapshots in Fig. 3a) until equilibration with the bath produces a self-trapped exciton in the middle of the oligomer. Remarkably, the same scenario holds for wavefunction evolution in other systems: we observe sloshing motions between two semi-circles in the nanohoop (Fig. 3b), right and left dendritic branches (Fig. 3c), and two monomers in a dimer (Fig. 3d and a Supplementary Movie 1). There are some obvious differences owing to the specific molecular structure. For example, in the dendrimer case, a large initial delocalization arises from the high density of coupled excited states (Frenkel excitons) accessed by the initial excitation combined with thermal fluctuations producing an ensemble of conformations. We notice a localization of transition density on the right branch even before the non-adiabatic transition, owing to the presence of the Pechukas force. The subsequent non-adiabatic electronic transitions, driven by strong coupling to high-frequency vibrational modes, quickly lead to the appearance of a spatially localized state in conjunction with electronic energy dissipation into nuclear motions scattered across the entire molecule. Moreover, in all of the systems (Fig. 3) we observe intermittent spatial delocalization of the electronic wavefunction along the trajectory, set by the interplay of electronic and vibronic couplings coexisting in a given system. Such delocalization first usually emerges at the moment of non-adiabatic

transition ($\Delta t = 0$) and further re-appears during the wavefunction motion in the middle between localization on "right" and "left" sides of the system. The dissipative processes (bath degrees of freedom) limit the number of such periodic events.

**Capturing periodic dynamical signatures**. The signatures of such concerted vibronic dynamics can be followed by analyzing common descriptors for both vibrational and electronic degrees of freedom. Bond-length alternation, BLA (see Methods) is a typical parameter for monitoring C–C stretches[52]. Fig. 3a displays periodic out-of-phase (with respect to left and right molecular halves) BLA variations in the linear oligomer. Alternatively, we can monitor displacements of the torsion angle on the top and bottom sides of the hoop, which also conveys out-of-phase vibration, as illustrated in Fig. 3b. Identical periodic dynamical signatures can be observed by following electronic degrees of freedom where spatial distribution of the state transition density is a good descriptor[53]. This is illustrated for the dendrimer (Fig. 3c) and dimer (Fig. 3d) in the evolution of the fraction of transition density contained in each branch or monomer, revealing oscillations associated with the changes in wavefunction localization. Other calculated variations of BLA, torsions, and transition densities are shown in Supplementary Figs. 4–7. Altogether, there is a consistent picture of coupled electron-nuclei dynamics modulated by specific vibrational excitations initiated by non-adiabatic transitions.

## Discussion

It is interesting to note that such concerted in-phase coherent vibronic dynamics is observed across the entire ensemble of trajectories with slow decay for well over 100 fs at room temperature for all considered systems and others[52,54], overcoming effects of thermal fluctuations, solvent viscosity, and disorder. We mention spectroscopic observations of "coherent phonons" persisting up to picoseconds (e.g., in the case of carbon nanotubes[11]), when the entire ensemble of molecules undergoes in-phase vibrational motion. While we discuss here only fast C–C stretching, slow torsions along the chain represent another structural motion coupled to the electronic system. By averaging over the C–C vibrations, one can inspect these slow recurring motions on the timescale of several picoseconds as illustrated in the case of the nanohoop (see Supplementary Fig. 7). An important spectroscopic observation is that the broad pulse may create coherences between electronic states in the initial condition[1,4–7,9,10]. These aspects invite further investigation by direct electronic dynamics modeling using advanced methodologies capable of describing interacting trajectories such as coherent Gaussian wavepacket approaches or multi-configurational methods[55,56].

In summary, we show the appearance of coherent electron-vibrational dynamics initiated by non-adiabatic transitions between excited states. Our concept is verified by direct atomistic NEXMD simulations of internal conversion in typical organic conjugated systems such as oligomer, hoop, dendrimer, and a molecular dimer. In all cases, we observe remarkably similar excited state dynamics initiated by non-adiabatic transitions between states leading to a specific asymmetric vibrational excitation, which modulates subsequent spatial evolution of the electronic wavefuntion described as wave-like motion. Consequently, we conclude that these phenomena are omnipresent across a very broad range of molecular materials and may potentially provide an alternative interpretation of existing and future spectroscopic experiments. Namely, an inevitable energy flow from electronic degrees of freedom to vibrations in the process of non-radiative relaxation and in the presence of strong electron–phonon coupling creates specific vibrational excitations that spatially modulate the excited electronic state before localizing it into a "self-trapped" excitation. Thus, there exists a dynamical

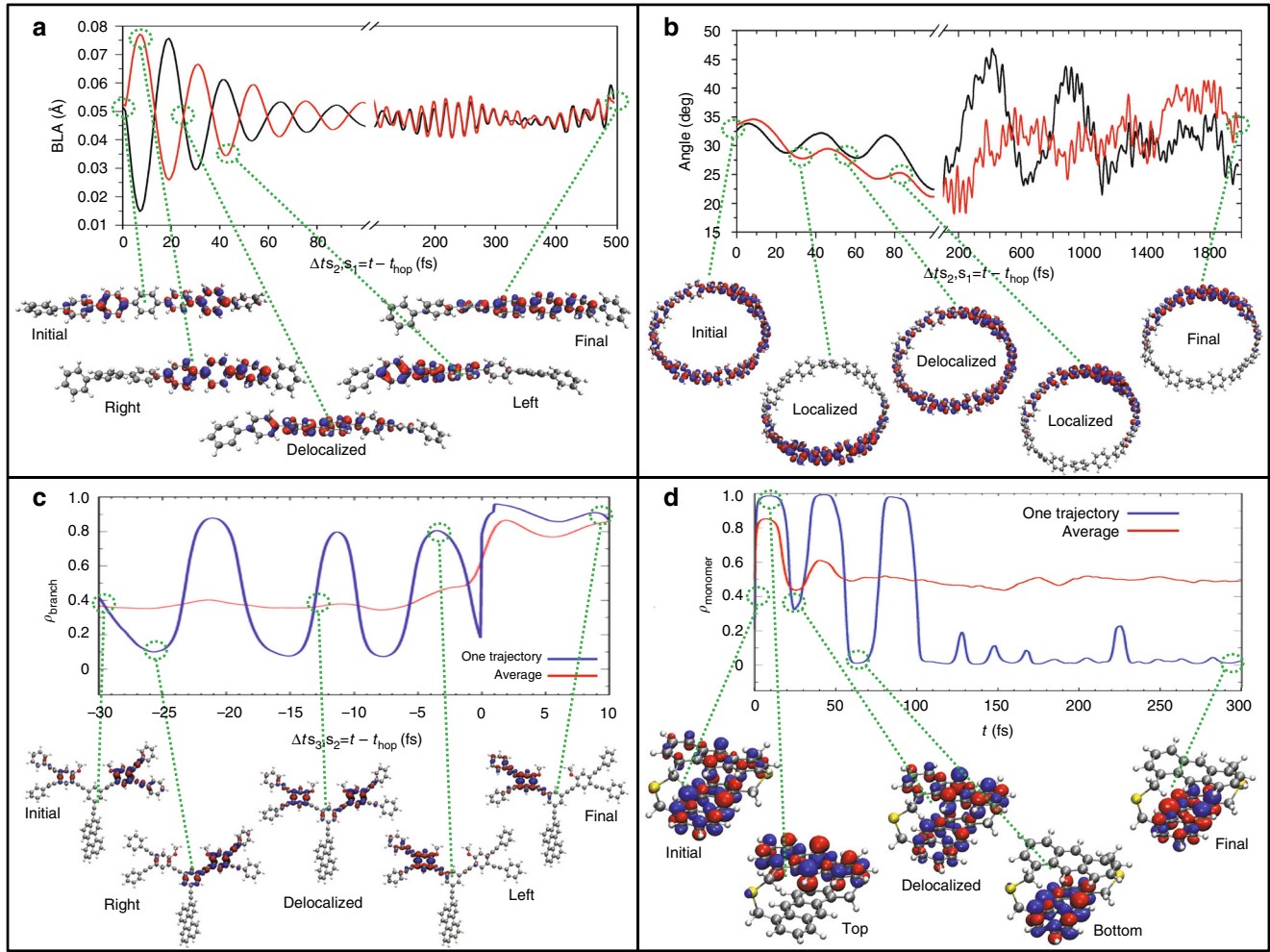

**Fig. 3** Electronic and vibrational dynamics of four model systems. **a–d** Correspond to the Oligomer, Nanohoop, Dendrimer, and Dimer, respectively, as presented in Fig. 2. In order to synchronize the NA transitions among individual trajectories, we introduce the convenient time variable $\Delta t = t - t_{hop}$. The exact moment of NA transition, which varies among trajectories, is set to $\Delta t = 0$. Negative values of $\Delta t$ correspond to times before the NA transition and positive values represent times after the transition. **a** The evolution of the average bond-length alternation, BLA, for the right (black) and left (red) molecular halves of the linear oligomer reveals out-of-phase oscillations corresponding to the change in localization between the two sides of the molecule confirmed by orbital plots of the transition density for snapshots taken during dynamics. **b** The evolution of the average benzene–benzene dihedral angle for the top (red) and bottom (black) molecular halves of the nanohoop indicate displacements of the ground state vibrational normal modes recovered after 200 fs. The corresponding orbital plots of the transition density for snapshots during dynamics reveal a transfer of wavefunction from bottom to top half of the nanohoop. **c** The evolution of the fraction of the transition density on the left branch of the dendrimer is plotted for a typical single trajectory (blue) and the ensemble average (red). The time axis denotes the time from the $S_3$ to $S_2$ transition (hop). Before the transition ($t - t_{hop} < 0$), the system on the upper state experiences oscillations between the two branches. After the transition to the lower state ($t - t_{hop} > 0$), the exciton becomes trapped in a single branch. The change in wavefunction localization is confirmed by orbital plots of the transition density during dynamics revealing transfer between left and right branches. **d** The evolution of the fraction of the transition density on the top monomer of the dendrimer is plotted for a typical single trajectory (blue) and the ensemble average (red). The oscillations correspond to the transfer between the top and bottom monomer shown in the orbital plots of the transition density taken from snapshots during dynamics

regime in which vibrations may efficiently transfer the electronic excitation across molecular constituents. Across all examples studied, such dynamics are vastly different from system to system in terms of persistence and timescales including cases of coupled multi-chromophore systems. Consequently, it may be possible to achieve the desired function (such as specific directed funneling of excitons) by relying on observed ultrafast dynamics of exciton-vibrations (e.g., by seeking a dynamical regime underpinning an efficient transport in multi-chromophore systems with large disorder and strong electron–phonon coupling). Thus, these observed underlying physical principles can be further exploited for design of functional organic materials for various optoelectronic applications.

## Methods

**Non-adiabatic excited state molecular dynamics.** The non-adiabatic excited-state molecular dynamics (NEXMD) software package[33] has been used to simulate the photoexcitation and subsequent electronic and vibrational energy relaxation and redistribution of each system: an anthracene dimer dithia-anthracenophane (DTA), a cycloparaphelynene with 16 phenyl units ([16]CPP), an unsymmetrical phenylene–ethynylene dendrimer with an ethynylene–perylene sink, and a linear paraphenylene with 7 phenyl units. The NEXMD combines the fewest switches surface hopping (FSSH) algorithm[57] with "on the fly" analytical calculations of excited-state energies[53,58,59], gradients[60,61], and non-adiabatic coupling terms[62–64]. The collective electronic oscillator (CEO) approach[65–67] is used to compute excited states at the configuration interaction singles (CIS) level of theory[68]. The semi-empirical AM1 Hamiltonian[69] has been used for all systems except for DTA where the PM3 Hamiltonian[70] is used.

**NEXMD simulation details and parameters**. One nanosecond ground state molecular dynamics simulations were performed for initial equilibration of all molecular structures studied. The Langevin thermostat[71] is used with temperature $T = 300$ K, a friction coefficient $\gamma = 20.0$ ps$^{-1}$ and time step $\Delta t = 0.5$ fs. The ground state trajectory was used to collect sets of initial configurations for the subsequent NEXMD simulations. The NEXMD simulations were started from these initial configurations by instantaneously promoting the system to an initial excited state $\alpha$ with the energy $\Omega_\alpha$, selected according to a Frank-Condon window defined as $g_\alpha = f_\alpha \exp\left[-T^2(E_{\text{laser}} - \Omega_\alpha)^2\right]$. $f_\alpha$ represents the normalized oscillator strength for the $\alpha$ state, and $E_{\text{laser}}$ represents the energy of a laser pulse centered at the maximum of the absorption spectra of a given molecule. The excitation energy width is given by the transform-limited relation of a Gaussian pulse with a full width half maximum (FWHM) of 100 fs, giving a value of $T^2 = 42.5$ fs. Using $g_\alpha$, the initial excited state for each equilibrated structure was determined.

Ten electronic excited states and their corresponding non-adiabatic couplings have been considered during NEXMD simulations for all systems. In agreement with previous numerical tests, 400 trajectories is found to be sufficient to achieve statistical convergence. A classical time step of 0.1 fs has been used for nuclear propagation and a quantum time step of 0.025 fs has been used to propagate the electronic degrees of freedom. Empirical corrections were introduced to account for electronic decoherence[72] and trivial unavoided crossings were diagnosed by tracking the identities of states[73]. The coherent vibronic dynamics observed in the present systems occur after the final effective hop to the lowest energy state and are therefore not an artifact of the decoherence model employed here[72]. Upon transition, the system decoheres instantaneously and moves independently on the lower surface with electron-vibrational coherent dynamics. In fact, the observed dynamics remains roughly the same if decoherence corrections are employed for the original FSSH method or not. These corrections primarily affect the relaxation timescales and eliminate numerical inconsistencies from the original FSSH[74]. More details concerning the NEXMD implementation and parameters can be found elsewhere[33,72,73,75].

**Analysis of electronic transition density**. During the NEXMD simulations, the electronic energy redistribution is monitored by computing the time-dependent localization of the electronic transition density, whose diagonal elements $(\rho^{g\alpha})_{nn}$ (index $n$ refers to atomic orbital (AO) basis functions) represent the changes in the distribution of the electronic density induced by photoexcitation from the ground state g to an excited electronic $\alpha$ state[76]. The orbital representation of the transition density is convenient for the analysis of excited state properties. For example, natural transition orbitals (NTOs)[77] enable the analysis of electron-hole separation in excitonic wavefunctions and charge transfer states by representing the electronic transition density matrix as essential pairs of particle and hole orbitals. Similarly, the orbital representation of the diagonal elements of the transition density is beneficial for the analysis of the total spatial extent of the excited state wavefunction. By partitioning the molecular system into moieties and/or chromophore units, the fraction of transition density, $(\rho^{g\alpha}(t))_X$, localized on each unit $X$ at a given time can be obtained by summing the contributions of the AO from each atom (index $A$) in $X$ and occasionally contributions of the AO from atoms localized on the boundary with another unit (index $B$)

$$\left(\rho^{g\alpha}(t)\right)_X^2 = \sum_{n_A m_A}\left(\rho^{g\alpha}_{n_A m_A}(t)\right)^2 + \frac{1}{2}\sum_{n_B m_B}\left(\rho^{g\alpha}_{n_B m_B}(t)\right)^2 \tag{1}$$

**Analysis of bond length alternation**. Molecular conformations during NEXMD simulations are analyzed by following the bond-length alternation (BLA). BLA and torsions (dihedral angles) represent the nuclear motions that are strongly coupled to the electronic degrees of freedom. BLA provides a convenient vibrational descriptor that reflects the inhomogeneity in the distribution of electrons along the $\pi$-conjugated molecule and it is generally defined as a difference between single and double bond lengths along the conjugated chain

$$\text{BLA} = d_1 - d_2 \cdot \frac{2}{3} - d_3 \cdot \frac{1}{3}, \tag{2}$$

where $d_1$, $d_2$, and $d_3$ are consecutive bond lengths in the conjugated system. Smaller values of BLA are associated with better $\pi$-conjugation and, therefore, an enhancement of the electronic delocalization[78,79]. Torsions are typically slower motions than C–C stretches. Here, the torsional motion of interest refers to the inter-ring dihedral angle, that indicates how rotated phenyl rings are with respect to a neighboring ring. The inter-ring dihedral angle modulates $\pi$-electron delocalization (large inter-ring dihedral angles can create conjugation breaks) and affects the molecular relaxation pathways.

**Data availability**. All relevant data are available from the authors upon request.

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

## Acknowledgements

This work was done in part at the Center for Integrated Nanotechnology (CINT), a U.S. Department of Energy, Office of Basic Energy Sciences user facility, and at the Center for Nonlinear Studies (CNLS) at Los Alamos National Laboratory (LANL). S.F.A. is supported by CONICET, UNQ, ANPCyT (PICT- 2014–2662). S.T. and T.N. acknowledge support from LANL Directed Research and Development Funds (LDRD). This research used resources provided by the LANL Institutional Computing (IC) Program. LANL is operated by Los Alamos National Security, LLC, for the National Nuclear Security Administration of the U.S. Department of Energy under contract DE-AC52-06NA25396.

## Author contributions

T.N. analyzed the data, prepared figures, and prepared the manuscript. D.O.-A., N.O., B.R.-H., L.A.-H., J.F.G. performed simulations of dendrimers, oligomers, dimers, and hoops and analyzed the data. V.D.K., S.F.-A., A.E.R., and S.T. designed the research and

devised the interpretation of the results. S.T. wrote the manuscript and all authors discussed the results and commented on the manuscript.

## Additional information

**Competing interests:** The authors declare no competing interests.

