## [Peer Review File · Nature Communications]

Reviewers' comments:

Reviewer #1 (Remarks to the Author):

This work provides a generalized description on the coherent electron-vibrational oscillation induced by non-adiabatic transition of various organic molecules. Authors claim that excess energy from the asymmetric-to-symmetric electronic transition is absorbed by the vibrational states, whose nuclear velocities match the non-adiabatic coupling (NAC) vector. This initiates the "sloshing" of wavefunctions between the molecular segments, which can be monitored by various parameters, such as bond length alternation, torsional angle, and distribution of transition density. Polyphenylene oligomer, cycloparaphenylene nanohoop, branched phenylene ethynylene dendrimer, and anthracene dimer were tested and all of the systems displayed asymmetric vibrational motion initiated by non-adiabatic transition, which supports authors' arguments. While the generality of the investigated excited-state dynamics is agreeable, I doubt if this has enough novelty which meets high standards of Nat. Commun. The described process has been discussed, in separate molecular systems, by the authors' group in previous works and this work seems like a collection of previously investigated systems. In this regard, this work is more like a short review rather than a communication and is more opt for more specific journals, such as J. Phys. Chem. Lett.

Reviewer #2 (Remarks to the Author):

RE: Coherent Exciton-Vibrational Dynamics and Energy Transfer in Conjugated Organics

Using an intuitively simple and general model to motivate their theory, the Authors propose that ultrafast "coherent" (wave-like) dynamics in "a broad range of molecular systems" arise from basic symmetry properties of electronic excited states and the non-adiabatic transitions that lead to internal conversion following photoexcitation. This proposal, pointing out that the non-adiabatic coupling vector necessarily initiates motion in "anti-symmetric" modes of the molecules that lead to "sloshing" of the electronic states, is backed up by numerical simulations of a number of different organic molecules, each with rather different topologies and modes of vibration. It is found that "coherent" dynamics appear following relaxation is almost all cases, often lasting on similar timescales (characteristic of c-c vibrations (fs) and torsions (ps)) in all systems. It is further proposed that this is a likely origin of many of the experimentally observed "coherent" phenomena in organics that have been attracting a great deal of interest across the physical sciences over the last few years.

At a first glance, the essential idea is elegant, broad and compelling, and the numerical results appear conclusively to confirm their arguments. Consequently, the work definitely appears - to me - to be highly suited, in scope and general presentation, for the wide readership of Nature Communications. However, after more detailed reading and consideration, I have a number of concerns about the claims and results in the paper that I think need to be clarified before I can recommend publication in Nature Communications.

Key questions:

1. The essential idea is that excited states in conjugated molecules can be loosely considered to have alternating symmetries (or, at least the lowest two can be thus considered), in real space (i.e. Ag & Bu, etc.). They then point out that the non-adiabatic coupling vector that drives transitions always has an "anti-symmetric" displacement pattern. I absolutely agree with this insight, but it wasn't clear to me from the MS if this result was being presented a new observation, or something already well-known (but of interest in this present context). For example, it is a

widely known selection rule that, given two electronic states of different "parity", the coupling operator in the matrix element between them must be anti-symmetric...this applies to light-matter interactions as much to vibronic coupling. For molecules possessing point group symmetries, similar selection rules for couplings are also well-known (this seems to be seen most clearly in the dendrimer example). Indeed, for the linear and circular chain systems, I'd also expect a discrete analogue of momentum conservation in the non-adiabatic transition, similar to electron-phonon scattering in the solid state. This isn't discussed, but presumably may be important in terms of the coupling to the vibrational modes supported in the structures and the energy gaps between molecules?

The physics is described uniquely in the language of the non-adiabatic molecular dynamics, which, here, is an appropriate method, but it seems to me that the basic physics is rather more general. As the problem of coherence, quantum effects, etc. in organic and biological systems is a highly interdisciplinary community, it would be very useful for the authors to describe the simple principle illustrated by the numerics in a more widely accessible way with appropriate references.

2. While the physical arguments given by the author are compelling, there wasn't - again, for me - enough detail about the numerical method for me to be completely satisfied that the generic phenomena seen in the examples are not due to the method employed. A detailed treatise on the method is not required, but I'd be grateful if the authors could answer the following in their reply:

a. The method is semiclassical, and the wave packet evolves on a given surface until it hops onto another, after which the system further evolves under the classical forces of the new PES. Is that it? My question is how this might relate to the more explicitly quantum mechanical treatments of "non-adiabatic coupling", or open quantum system theory (normally performed in a diabatic basis), where these transitions might be thought of in terms of (incoherent) emission of single, or a few quanta. In several of the systems mentioned by the authors, particularly photosynthetic complexes, studies of coherence dynamics use this approach, normally with advanced techniques that improve upon the Redfield approximation of open quantum dynamics (see some suggested references, below). A single quanta in a mode does not cause the mode to be displaced, which leads to my next question.

b. In NEXMD, following an electronic transition, the excess energy is "redistributed" among nuclear coordinates - is this random, in the sense of the excess energy appearing (after ensemble averaging) as heat? I ask, as the very nice average oscillations seen in the BLA and dihedral angles seem to have a very definite phase, but how does this come about, if the motions along the coupling vector are random in each instance? Similarly, are the "snap shot" transition densities shown in Fig. 3a & b individual trajectories or averaged values? It would be useful for the reader to indicate the times at which these snap shots were taken (on the post-hop timescale showing the oscillations, as in Fig 3c). In the case of the dendrimer, why does the ensemble (thermal?) distribution break symmetry and become trapped at a particular branch (left) at long times?

c. The methods section mentions that some processing is made to handle decoherence and dephasing, which are very important in the case of fully quantum methods of vibronic dissipation. How are coherences handled, and what is the effective dephasing time? Is it comparable to the ~100fs during which the early oscillatory behaviours are observed?

d. The subsequent oscillatory motion of the electronic density in the molecules is a direct consequence of motion on the lower adiabatic surfaces, which must show large changes in localisation along the vibrational modes of the lower PES that are excited by the NA coupling. The authors mention that the displacement experienced by the molecule in the lower PES is not a normal mode but a wave packet cluster of modes. Is there any sense of the spectral width of this wave packet in the numerical examples, again, is the width comparable to any of the dynamical timescales observed in the numerical results?

e. Finally, if the energy separations between excited states are comparable to the frequencies of intramolecular motions, as they may in long polymers or pigment-protein complexes, is the adiabatic (born-oppenheimer) approximation a reasonable framework for the dynamics? The manuscript and SI do not give any details about the energy levels, energy gaps at the ground state geometry, etc. It would be very helpful to include these.

3. a. Initial conditions and excitation. Again, the numerical results are presented without much explanation of the initial conditions (not until the methods section). From what I was able to understand, a high lying excited state is occupied by a laser pulse of 100fs duration. Which? The time axis in the figures is the time after the system has hopped to the "S2" state? Are all ensemble averages taking over this time variable, rather than the absolute time from excitation? This may be relevant, and should be clarified.

No actual values are given for the energy of the excitation; this should be presented, along with the energy levels of the 10 excited states consider per systems. This is important for another reason, the subject of considerable debate in photosynthetic systems: does the width of the pulse create coherences between electronic states in the initial condition (i.e. is the bandwidth broad enough to excite superpositions of levels, including the two that are investigated in detail)? If so, how does this effect the results obtained by NEXMD?

Presentation issues:

1. The manuscript begins with a definition of coherence which is rather vague. Off diagonal elements in a density matrix can depend (trivially) on a choice of basis, so the authors need to clarify precisely what they define as coherence (spatial, between eigenstates of some Hamiltonian...). For example, from the point that the "anti-symmetric" motion has been created on the lower PES and is out of the Na region, the dynamics of the system ("the sloshing") are essentially classical motions on an adiabatic surface, which for quantised nuclear degrees of freedom is a type of "coherent" motion, but one that is as close as possible to classical (this is the approximation used in the numerical method). "Coherence" of this kind, as might be generated by raman scattering onto a ground state wave packet can often persist on timescales $\gg 10$ -100fs at RT. A paper making a similar point to this (rapidly changing electronic character along a co-ordinate coupling two low lying states) is that of Tiwari 2013 (see below).

The BO approximation also means that no entanglement between the electronic and bath degrees of freedom exist, so care must be taken in the definition of "coherence". Many of the works involving quantum baths place considerable emphasis on the establishment of quantum correlations between the system and bath, which cannot be described in a semiclassical method.

2. The introduction begins by highlighting the "discovery" of coherent behaviour in photosynthetic systems (~2007) and later mentions that "oscillatory evolution" of wavefunctions can improve transport. They go on to discuss the change in thinking, from purely electronic coherences to a more complex interplay of electronic and vibrations processes. For the latter point of view, the authors point to [1,3], both from 2017, missing an enormous body of literature on this subject, particularly in photosynthesis, that made precisely this point (normally within models that explicitly treat the vibrations as quantum degrees of freedom). Some of the most highly cited works include the following (but there are many more excellent works on this subject, it is still an active subject)

@article{christensson2012origin,
title={Origin of long-lived coherences in light-harvesting complexes},

author={Christensson, Niklas and Kauffmann, Harald F and Pullerits, Tonu and Mančal, Tomáš},
journal={The Journal of Physical Chemistry B},
volume={116},
number={25},
pages={7449--7454},
year={2012},
publisher={ACS Publications}
}

@article{tiwari2013electronic,
title={Electronic resonance with anticorrelated pigment vibrations drives photosynthetic energy transfer outside the adiabatic framework},
author={Tiwari, Vivek and Peters, William K and Jonas, David M},
journal={Proceedings of the National Academy of Sciences},
volume={110},
number={4},
pages={1203--1208},
year={2013},
publisher={National Acad Sciences}
}

@article{kolli2012fundamental,
title={The fundamental role of quantized vibrations in coherent light harvesting by cryptophyte algae},
author={Kolli, Avinash and O'Reilly, Edward J and Scholes, Gregory D and Olaya-Castro, Alexandra},
journal={The Journal of chemical physics},
volume={137},
number={17},
pages={174109},
year={2012},
publisher={AIP}
}

@article{chin2013role,
title={The role of non-equilibrium vibrational structures in electronic coherence and recoherence in pigment--protein complexes},
author={Chin, AW and Prior, J and Rosenbach, R and Caycedo-Soler, F and Huelga, SF and Plenio, MB},
journal={Nature Physics},
volume={9},
number={2},
pages={113},
year={2013},
publisher={Nature Publishing Group}
}

@article{kreisbeck2012long,
title={Long-lived electronic coherence in dissipative exciton dynamics of light-harvesting complexes},
author={Kreisbeck, Christoph and Kramer, Tobias},

```
journal={The Journal of Physical Chemistry Letters},
volume={3},
number={19},
pages={2828--2833},
year={2012},
publisher={ACS Publications}
}
```

```
@article{falke2014coherent,
title={Coherent ultrafast charge transfer in an organic photovoltaic blend},
author={Falke, Sarah Maria and Rozzi, Carlo Andrea and Brida, Daniele and Maiuri, Margherita and Amato, Michele and Sommer, Ephraim and De Sio, Antonietta and Rubio, Angel and Cerullo, Giulio and Molinari, Elisa and others},
journal={Science},
volume={344},
number={6187},
pages={1001--1005},
year={2014},
publisher={American Association for the Advancement of Science}
}
```

```
@article{bakulin2016real,
title={Real-time observation of multiexcitonic states in ultrafast singlet fission using coherent 2D electronic spectroscopy},
author={Bakulin, Artem A and Morgan, Sarah E and Kehoe, Tom B and Wilson, Mark WB and Chin, Alex W and Zigmantas, Donatas and Egorova, Dussia and Rao, Akshay},
journal={Nature chemistry},
volume={8},
number={1},
pages={16},
year={2016},
publisher={Nature Publishing Group}
}
```

```
@article{lim2015vibronic,
title={Vibronic origin of long-lived coherence in an artificial molecular light harvester},
author={Lim, James and Paley, David and Caycedo-Soler, Felipe and Lincoln, Craig N and Prior, Javier and Von Berlepsch, Hans and Huelga, Susana F and Plenio, Martin B and Zigmantas, Donatas and Hauer, Jürgen},
journal={Nature communications},
volume={6},
pages={ncomms8755},
year={2015},
publisher={Nature Publishing Group}
}
```

3. The authors conclude that the insights arising from this work could be "exploited for design of functional organic materials". Could the authors please give one or two (brief, concrete) examples? Could the authors comment on how these results might translate to the case of nanoscale supramolecular complexes (most application of "coherence" are envisioned in systems containing multiple, not single, organic molecules)? Is there - generally - an obvious co-ordinate or adiabatic PES for a well-separated multi-chromophore system?

In conclusion, I believe that the fundamental idea proposed in this paper is correct and of

potentially high impact in this field, but revisions to this manuscript and a detailed response to my questions are required before I can recommend it for publication.

Reviewer #1 (Remarks to the Author):

This work provides a generalized description on the coherent electron-vibrational oscillation induced by non-adiabatic transition of various organic molecules. Authors claim that excess energy from the asymmetric-to-symmetric electronic transition is absorbed by the vibrational states, whose nuclear velocities match the non-adiabatic coupling (NAC) vector. This initiates the “sloshing” of wavefunctions between the molecular segments, which can be monitored by various parameters, such as bond length alternation, torsional angle, and distribution of transition density. Polyphenylene oligomer, cycloparaphenylene nanohoop, branched phenylene ethynylene dendrimer, and anthracene dimer were tested and all of the systems displayed asymmetric vibrational motion initiated by non-adiabatic transition, which supports authors’ arguments. While the generality of the investigated excited-state dynamics is agreeable, I doubt if this has enough novelty which meets high standards of Nat. Commun. The described process has been discussed, in separate molecular systems, by the authors' group in previous works and this work seems like a collection of previously investigated systems. In this regard, this work is more like a short review rather than a communication and is more opt for more specific journals, such as J. Phys. Chem. Lett.

We thank the Reviewer for the kind assessment of our manuscript. We humbly disagree that the present work is merely a review of our previous work. It is true that the systems presented here have been featured in previously published works. However, the present study is not simply a review of the electronic dynamics from our previous work. Instead, our article introduces a new theory of coherent dynamics, and presents a model of wave function symmetry as related to non-adiabatic transitions during internal conversion. This work is the first time that the idea is presented: anti-symmetric motions of molecules are activated through the non-adiabatic coupling vector to produce "sloshing" of the electronic wavefunction between molecular segments. We use very different molecular classes, to support our model and demonstrate its generality in a broad range of systems and indeed, the data shown in the present work has not been previously published.

Again, the theory presented is being introduced here for the first time. Our work shows that the asymmetric shape of the electronic couplings between states induces asymmetric vibrational coherences that lead to wave-like spatial oscillations of the electronic transition density. This sloshing is the result of exciton-vibrational coherences induced by the internal conversion process.

Reviewer 2 agrees that the “fundamental idea proposed in this paper is correct and of potentially high impact in this field”. We believe the universal mechanism presented in this paper represents a paradigm change regarding the process by which light-induced coherence forms in molecular materials and the interpretation of spectroscopic data. This concept will have far-reaching impacts for scientists working on biological systems, light-harvesting systems, materials science, and beyond. Therefore, we believe that our work is of broad interest and high impact, making it well suited for publication in Nature Communications.

In response to your concerns and multiple questions of Referee #2, upon revision we present the finding of the MS in more general context, as outlined in the details below.

Reviewer #2 (Remarks to the Author):

RE: Coherent Exciton-Vibrational Dynamics and Energy Transfer in Conjugated Organics

Using an intuitively simple and general model to motivate their theory, the Authors propose that ultrafast "coherent" (wave-like) dynamics in "a broad range of molecular systems" arise from basic symmetry properties of electronic excited states and the non-adiabatic transitions that lead to internal conversion following photoexcitation. This proposal, pointing out that the non-adiabatic coupling vector necessarily initiates motion in "anti-symmetric" modes of the molecules that lead to "sloshing" of the electronic states, is backed up by numerical simulations of a number of different organic molecules, each with rather different topologies and modes of vibration. It is found that "coherent" dynamics appear following relaxation in almost all cases, often lasting on similar timescales (characteristic of c-c vibrations (fs) and torsions (ps)) in all systems. It is further proposed that this is a likely origin of many of the experimentally observed "coherent" phenomena in organics that have been attracting a great deal of interest across the physical sciences over the last few years.

At a first glance, the essential idea is elegant, broad and compelling, and the numerical results appear conclusively to confirm their arguments. Consequently, the work definitely appears - to me - to be highly suited, in scope and general presentation, for the wide readership of Nature Communications. However, after more detailed reading and consideration, I have a number of concerns about the claims and results in the paper that I think need to be clarified before I can recommend publication in Nature Communications.

We would first like to thank the Reviewer for the very thorough summary and thought-provoking analysis of our manuscript. We are very pleased that the Reviewer recognizes the significance and potential impact of our work and appreciate the comments and questions that have helped us to improve the revised manuscript.

Key questions:

1. The essential idea is that excited states in conjugated molecules can be loosely considered to have alternating symmetries (or, at least the lowest two can be thus considered), in real space (i.e. Ag & Bu, etc.). They then point out that the non-adiabatic coupling vector that drives transitions always has an "anti-symmetric" displacement pattern. I absolutely agree with this insight, but it wasn't clear to me from the MS if this result was being presented as a new observation, or something already well-known (but of interest in this present context). For example, it is a widely known selection rule that, given two electronic states of different "parity", the coupling operator in the matrix element between them must be anti-symmetric...this applies to light-matter interactions as much to vibronic coupling. For molecules possessing point group symmetries, similar selection rules for couplings are also well-known (this seems to be seen most clearly in the dendrimer example). Indeed, for the linear and circular chain systems, I'd also expect a discrete analogue of momentum conservation in the non-adiabatic transition, similar to electron-phonon scattering in the solid state. This isn't discussed, but presumably may be important in terms of the coupling to the vibrational modes supported in the structures and the energy gaps between molecules?

The physics is described uniquely in the language of the non-adiabatic molecular dynamics, which, here,

is an appropriate method, but it seems to me that the basic physics is rather more general. As the problem of coherence, quantum effects, etc. in organic and biological systems is a highly interdisciplinary community, it would be very useful for the authors to describe the simple principle illustrated by the numerics in a more widely accessible way with appropriate references.

The Referee is making excellent suggestions. First, we have put a discussion of non-adiabatic dynamics in a more general basic physics context in the revised MS as follows:

“Most of the systems studied above belong to the ‘intermediate coupling regime’, when the electronic and vibrational couplings are comparable¹³. The transport processes following photoexcitation are concomitant to non-radiative relaxation, when the system dissipates the excess of electronic energy into heat. During this internal conversion, energy typically flows from the electronic to vibrational degrees of freedom via two distinct mechanisms. When electronic states are well separated, the system can relax adiabatically downhill on a single potential energy surface within the Born-Oppenheimer framework. Alternatively, when electronic states are close in energy, the Born-Oppenheimer approximation breaks down and non-adiabatic evolution takes place when the electronic state (and the respective potential energy surface) changes during the dynamics^{3,4}. This is a common scenario for energy transfer. Here, one extreme includes strong electronic couplings...”

Moreover, to address the point whether the concepts of alternating state symmetries is ‘a new observation, or something already well-known (but of interest in this present context)’, we have added references and modified the text as follows:

“Here, we show how coherent exciton-vibrational dynamics emerges in photoactive molecular systems due to non-adiabatic (non-Born-Oppenheimer) transitions between excited states. Previous studies recognized the importance of symmetry of vibronic coupling between different electronic states in resonant transitions^{29,30}, electron^{31,32} and energy^{33,34} transfer rates. Here we are exploring its effect on coherent electron-vibrational dynamics. This phenomenon is ubiquitous...”

2. While the physical arguments given by the author are compelling, there wasn't - again, for me - enough detail about the numerical method for me to be completely satisfied that the generic phenomena seen in the examples are not due to the method employed. A detailed treatise on the method is not required, but I'd be grateful if the authors could answer the following in their reply:

a. The method is semiclassical, and the wave packet evolves on a given surface until it hops onto another, after which the system further evolves under the classical forces of the new PES. Is that it? My question is how this might relate to the more explicitly quantum mechanical treatments of "non-adiabatic coupling", or open quantum system theory (normally performed in a diabatic basis), where these transitions might be thought of in terms of (incoherent) emission of single, or a few quanta. In several of the systems mentioned by the authors, particularly photosynthetic complexes, studies of coherence dynamics use this approach, normally with advanced techniques that improve upon the Redfield approximation of open quantum dynamics (see some suggested references, below). A single quanta in a mode does not cause the mode to be displaced, which leads to my next question.

First, the Reviewer is correct that for each trajectory in the ensemble, the system always evolves under the classical forces of a single PES until it hops to a new state.

Also the Reviewer raises then a very interesting distinction. In semi-classical non-adiabatic MD, we are interested in the explicit correlation between the motion of the nuclei and the electronic dynamics, plus we treat the nuclei to all orders of anharmonicity, but pay the price of treating them classically.

In Open quantum systems, nuclei are usually treated as an effective bath and the self-energy due to coupling of the nuclei and the electrons is usually defined in frequency space effectively time averaging over the nuclear motion and losing the explicit correlation, not good for describing chemistry, isomerization, etc. The interaction is typically perturbative, to second order (Redfield) or more advanced (Non-Crossing Approximation, CTMA). The self-energy can be described using various models, and non-adiabatic coupling can be calculated along adiabatic, mean-field or surface hopping dynamics and Fourier transform to get a self-energy but usually the coupling strength is just estimated rather than being ab-initio. The perturbative approach does not lead to the "decoherence" problem found in surface hopping, however nuclear dynamics are still averaged, and assumed not to undergo large change. Redfield is very similar to Ehrenfest, if only the electronic coefficients were followed, but with a decoherence correction from the perturbative treatment of the electron-vibration interaction, and of course averaging over the vibrational dynamics entirely, making the PES constant.

Of course the system could be defined fully quantum, i.e. in terms of vibronic states. However, this approach is extremely numerically expensive for more than one oscillator (i.e., more than 1D), just like solving the full electron-nuclear dynamics. In some cases, the many-phonon systems can be solved exactly, but only for harmonic oscillators and linear coupling. That model is only for the electronic dynamics and does not give a picture of the coupled nuclear dynamics.

To put it more succinctly, the semiclassical method is more appropriate when the dynamics are far from ground state equilibrium, where the normal-modes would become a bad basis for quantum dynamics. It also has the added benefit that the electron-vibration self-energy does not have to be estimated, the interaction is explicit from the non-adiabatic coupling in real time.

To reflect the above discussion, we introduced a brief paragraph in the revised MS along with appropriate references:

"In a typical scenario for internal conversion (Fig. 1a), a photoexcited wave packet goes through the crossing region to transition from the upper to the lower PESs. Such processes are usually described via semiclassical models establishing consistent propagation of quantum (electrons) and classical (nuclei) degrees of freedom in the non-adiabatic regime. Ehrenfest and surface hopping are examples of such methods allowing explicit treatment of large molecular systems for which fully quantum dynamics is prohibitively expensive^{4,39-41}. Alternative perturbative approaches⁴²⁻⁴⁴ usually treat nuclei as an effective bath, and the self-energy due to coupling of the nuclei and electrons is usually defined in frequency space and is estimated by averaging over the nuclear motion, thus losing the explicit correlation. Such approaches have been extensively applied, for example, to biological light harvesting systems^{45,46}. In this study, instead, the correlation between the electronic and nuclear dynamics is explicitly included in real time through the non-adiabatic coupling."

b. In NEXMD, following an electronic transition, the excess energy is "redistributed" among nuclear coordinates - is this random, in the sense of the excess energy appearing (after ensemble averaging) as heat? I ask, as the very nice average oscillations seen in the BLA and dihedral angles seem to have a very definite phase, but how does this come about, if the motions along the coupling vector are random in each instance? Similarly, are the "snap shot" transition densities shown in Fig. 3a & b individual trajectories or averaged values? It would be useful for the reader to indicate the times at which these snap shots were taken (on the post-hop timescale showing the oscillations, as in Fig 3 c). In the case of the dendrimer, why does the ensemble (thermal?) distribution break symmetry and become trapped at a particular branch (left) at long times?

In NEXMD, the dissipated electronic energy is not redistributed randomly for every single trajectory. It is instead distributed in the direction of the non-adiabatic coupling vector upon non-adiabatic transition. The direction of this vector is highly significant and it represents the direction of the driving force propagating the instantaneous molecular configurations throughout regions of strong coupling. This is a common feature for all semiclassical methodologies, beyond the surface hopping algorithm we use (A. J. White, S. Tretiak, and D. Mozyrsky, Coupled wave-packets for non-adiabatic molecular dynamics: a generalization of Gaussian wave-packet dynamics to multiple potential energy surfaces, *Chem. Sci.*, 7, 4905 (2016)). We have previously shown that the non-adiabatic coupling vector is related to specific excited state normal modes (Soler, M.A., Roitberg, A.E., Nelson, T., Tretiak, S. & Fernandez-Alberti, S. Analysis of State-Specific Vibrations Coupled to the Unidirectional Energy Transfer in Conjugated Dendrimers. *J. Phys. Chem. A*, 116, 9802–9810 (2012); Soler, M.A., Nelson, T., Roitberg, A.E., Tretiak, S. & Fernandez-Alberti, S. Signature of Nonadiabatic Coupling in Excited-State Vibrational Modes. *J. Phys. Chem. A*, 118, 10372–10379 (2014)) such that the driving force acts along a unique normal mode direction. This provides a simple physical rationale for adjusting nuclear velocities along the direction of the non-adiabatic coupling vector following hops between electronic states. Moreover, the fact that the direction of NAC vector defines the flux of energy toward specific vibrations has been emphasized by Bittner et al. (Yang, X., Keane, T., Delor, M., Meijer, A.J.H.M., Weinstein, J. & Bittner, E.R. Identifying electron transfer coordinates in donor-bridge-acceptor systems using mode projection analysis, *Nature Comm.*, 8, 14554 (2017); Yang, X. & Bittner, E.R. Intramolecular charge- and energy- transfer rates with reduced modes: comparison to Marcus theory for donor-bridge-acceptor systems, *J. Phys. Chem. A*, 118, 5196 (2014)) in the case of charge transfer by proposing an efficient Lanczos algorithm.

The transition density snap shots shown in Figure 3a and b are from individual trajectories whose dynamics are representative of the average behavior observed for the ensemble. In the revised manuscript, we have indicated the times corresponding to the snap shots in Figures 3a and b. We observe well defined oscillatory motion for every individual trajectory in the ensemble for every system considered. This oscillatory motion is also present albeit on the shorter time scales across the entire ensembles of trajectories. On page 9 of the manuscript we attribute such collective concerted oscillations to the coherent phonons previously experimentally observed, for example, in carbon nanotubes.

For the dendrimer, localization is happening spontaneously either to the left or right branch for different trajectories as expected. Subsequently, we should first clarify that the branches

are indistinguishable from one trajectory to the next. That is, whichever branch has the highest transition density is treated as the localized branch. But from one trajectory to another, they are not necessarily the same. That depends on the conformational fluctuations that order the excited states. The initial S_3 excited state is delocalized throughout the backbone, while after the transition to S_2 , it becomes localized on a single branch, through coupling with nuclear dynamics. The ultrafast localization to a single branch occurs due to the non-adiabatic transition. Specifically, nuclear dynamics on different excited state surfaces are responsible for the ultrafast transient localization (i.e., self-trapping) of the electronic transition density. This localization of the excited-state wave function occurs on an ultrafast (hundreds of femtoseconds) time scale due to non-adiabatic transitions between excited states. For this dendrimer, the large initial delocalization is due to the high density of coupled excited states (Frenkel excitons) accessed by the initial excitation and the thermal fluctuations that yield an ensemble of structures with varying conformations. The non-adiabatic electronic transitions, driven by strong coupling to high-frequency vibrational modes, quickly leads to the appearance of a spatially localized intermediate state with concomitant conversion of excess electronic energy into nuclear motions scattered across the entire molecule.

The above discussion is reflected in the following modifications in the revised MS:

“Furthermore, upon non-adiabatic transition, the excess electronic energy is dispersed into the nuclear velocities in the direction of the NAC vector to enforce energy conservation. The direction of the NAC vector is highly significant and it represents the direction of the driving force acting along a unique normal mode direction throughout regions of strong coupling^{48,49}. The fact that the direction of NAC vector defines the flux of energy toward specific vibrations has been emphasized by Bittner et al.^{50,51}. This provides a simple physical rationale for adjusting nuclear velocities along the direction of the non-adiabatic coupling vector. These electronic-to-vibrational energy conversion principles were proven at various levels of theory.”

To address the Reviewer’s question regarding the localization in dendrimer system, we have expanded our discussion in the revised MS:

“For example, in the dendrimer case, a large initial delocalization arises from the high density of coupled excited states (Frenkel excitons) accessed by the initial excitation combined with thermal fluctuations producing an ensemble of conformations. We notice a localization of transition density on the right branch even before the non-adiabatic transition, owing to the presence of the Pechukas force. The subsequent non-adiabatic electronic transitions, driven by strong coupling to high-frequency vibrational modes, quickly lead to the appearance of a spatially localized state in conjunction with electronic energy dissipation into nuclear motions scattered across the entire molecule.”

c. The methods section mentions that some processing is made to handle decoherence and dephasing, which are very important in the case of fully quantum methods of vibronic dissipation. How are coherences handled, and what is the effective dephasing time? Is it comparable to the ~ 100 fs during which the early oscillatory behaviours are observed?

The decoherence is treated with an instantaneous decoherence model. In this sense, there is no explicit dephasing time, rather dephasing is instantaneous. It is based on the assumption that following a hop, divergent wave packets will instantaneously separate in phase space and

immediately undergo independent evolution. In that sense, the wave packet undergoes fully coherent evolution between hops. See our previous work on the subject comparing a variety of existing empirical decoherence corrections (Nelson, T., Fernandez-Alberti, S., Roitberg, A. E. & Tretaik, S. Nonadiabatic Excited-State Molecular Dynamics: Treatment of Electronic Decoherence. *J. Chem. Phys.*, 138, 224111 (2013))

The coherent vibronic dynamics observed in the present systems occur *after* the final effective hop to S_1 , a lowest energy state. Upon transition, the system moves on S_1 independently of S_2 and dynamics on S_1 are fully coherent. Therefore, the oscillatory motions are not an artifact of the decoherence model. In fact, the observed dynamics remains roughly the same if decoherence corrections are employed for Tully's method or not. These primarily affect the relaxation timescales and eliminate numerical inconsistencies from the original FSSH (Hammes-Schiffer et al. Improvement of the Internal Consistency in trajectory surface hopping. *J. Phys. Chem. A.*, 1999, 103, 9399).

Some confusion may have been caused by our use of the word "dephasing" to describe the decay of coherent vibronic dynamics. We have clarified our statement in the revised MS as follows:

"It is interesting to note that such concerted in-phase coherent vibronic dynamics is observed across the entire ensemble of trajectories with slow decay for well over 100 femtoseconds at room temperature for all considered systems"

We have also included the following discussion of the dephasing model in the revised MS:

"The coherent vibronic dynamics observed in the present systems occur *after* the final effective hop to the lowest energy state and are therefore not an artifact of the decoherence model employed here⁸⁰. Upon transition, the system decoheres instantaneously and moves independently on the lower surface with electron-vibrational coherent dynamics. In fact, the observed dynamics remains roughly the same if decoherence corrections are employed for the original FSSH method or not. These corrections primarily affect the relaxation timescales and eliminate numerical inconsistencies from the original FSSH⁸²."

d. The subsequent oscillatory motion of the electronic density in the molecules is a direct consequence of motion on the lower adiabatic surfaces, which must show large changes in localisation along the vibrational modes of the lower PES that are excited by the NA coupling. The authors mention that the displacement experienced by the molecule in the lower PES is not a normal mode but a wave packet cluster of modes. Is there any sense of the spectral width of this wave packet in the numerical examples, again, is the width comparable to any of the dynamical timescales observed in the numerical results?

This is another excellent question of the Referee. The non-adiabatic coupling vector developed on the basis of the lower PES normal modes is commonly spread among a small subset of normal modes typically strongly coupled to the electronic degrees of freedom, which may have very different frequencies such as C-C stretches and torsions in the case of CPPs. A typical spectral width within each subclass of modes is less than 0.05 eV. These modes become active experiencing a substantial increase in their vibrational energy during the process (Shenai, P.M., Fernandez-Alberti, S., Bricker, W.P., Tretiak, S. & Zhao, Y. Internal Conversion and Vibrational Energy Redistribution in Chlorophyll A *J. Phys. Chem. B* 120(1), 49-58 (2016)). In another

example of our previous work on CPPs (N. Oldani, S. K. Doorn, S. Tretiak, and S. Fernandez-Alberti, Photoinduced dynamics in cycloparaphenylenes: planarization, electron-phonon coupling, localization and intra-ring migration of the electronic excitation. *Phys. Chem. Chem. Phys.*, 19, 30914-30924 (2017)), the projection of the non-adiabatic coupling vector on the basis of excited-states equilibrium normal modes calculated on S_1 reveals the main contributions of two normal modes with frequencies 1766.7 cm^{-1} and 1776.7 cm^{-1} respectively. These modes correspond to equivalent motions associated to in-plane E_{2g} vibrations of benzene but involving different phenyl units on the ring.

The above discussion is reflected in the following modifications in the revised MS:

“We expect that such vibrational excitation is related to the structural motions usually considered to be coupled to the electronic degrees of freedom such as C-C stretches and torsional librations. However, it is not directly associated with any of the vibrational normal modes of either the ground or any excited state, rather being a complex superposition of several normal modes, as was demonstrated in the case of charge transfer⁵³. In the present examples, the non-adiabatic coupling vector is commonly spread among a small subset of normal modes ($\sim 2-5$) such as C-C stretches and torsions. A typical spectral width within each subclass of modes is less than 0.05 eV. These modes become active experiencing a substantial increase in their vibrational energy during the process⁵⁴.”

e. Finally, if the energy separations between excited states are comparable to the frequencies of intramolecular motions, as they may in long polymers or pigment-protein complexes, is the adiabatic (Born-Oppenheimer) approximation a reasonable framework for the dynamics? The manuscript and SI do not give any details about the energy levels, energy gaps at the ground state geometry, etc. It would be very helpful to include these.

We fully agree with the Referee, and there is no assumption in the examples of the calculations that the dynamics is adiabatic. In fact, we continue running the FSSH non-adiabatic simulations with the NEXMD framework for the entire duration of the dynamics. For some systems, the wave packet passes through the non-adiabatic region fast (e.g., oligomer) and the subsequent dynamics can be denoted as essentially adiabatic, whereas for others (such as dendrimer or a dimer) represent the case when “the energy separations between excited states are comparable to the frequencies of intramolecular motions”. Here the non-adiabatic dynamics can persist for longer timescales. Subsequently we are not making any superficial distinction between adiabatic and non-adiabatic dynamics, which is always a system-specific feature and can be tuned (see our answer to the very last suggestion).

In the revised manuscript, we have added plots of the distribution of S_2-S_1 (S_3-S_2 for dendrimer) energy gaps after the effective hops to the SI. It is important to stress that the energy gaps between states are subject to change during the electronic relaxation after the initial excitation. If the electronic relaxation pathways lead the system to the conical intersection or their immediate regions where electronic states are not well separated and the Born-Oppenheimer description is not sufficient, values of the energy gaps lower than ~ 0.1 eV are expected. The new plots show that for the dendrimer, nanohoop, and dimer at the moment of the non-adiabatic transition ($\Delta t=0$), the energy gap is small and the distribution is very narrow (all trajectories exhibit a small energy gap). Average values of the energy gaps are $\Delta E(S_3-S_2) \sim$

0.04 eV, $\Delta E(S_2-S_1) \sim 0.11$ eV, and $\Delta E(S_2-S_1) \sim 0.06$, respectively. Therefore, adiabatic dynamics are not sufficient in this case.

To answer to this concern, we have added the following discussion to the manuscript and the new Figure to SI:

“Having confirmed the foundations of our model, we further turn to the analysis of dynamical variables. Surface hopping algorithms underpinning the NEXMD simulations produce a trajectory ensemble. The meaningful observables such as relaxation timescales, are then calculated as statistical averages. When non-radiative relaxation pathways lead the system to the regions where electronic states are not well separated and the Born-Oppenheimer (adiabatic) description is insufficient, values of the energy gaps lower than ~ 0.1 eV are expected. Plots of the relevant energy gap distributions for all systems (Fig. S2 in SI) confirm that near the non-adiabatic transitions, the energy gap is small with a narrow distribution across the ensemble. Consequently, no superficial distinction between adiabatic and non-adiabatic regimes is made in our simulations, and the NEXMD trajectories are run for the entire duration of the dynamics starting from a ground state configuration instantaneously promoted to an excited state (Fig. S3). For some systems, the wave packet passes through the non-adiabatic region fast (e.g., oligomer) and the subsequent dynamics is essentially adiabatic. However, others (such as dendrimer or dimer) represent the case when the energy separation between excited states is comparable to the frequency of intramolecular motions and non-adiabatic dynamics persist for longer timescales.”

Figure S2 Plots of the distribution of S_2-S_1 (S_3-S_2 for dendrimer) energy gaps after the effective hops. For the dendrimer, nano hoop, and dimer at the moment of the non-adiabatic transition ($\Delta t=0$), the energy gap is small and the distribution is very narrow (i.e., all trajectories exhibit a

small energy gap). Average values of the energy gaps are $\Delta E(S_3-S_2) \sim 0.04$ eV, $\Delta E(S_2-S_1) \sim 0.11$ eV, and $\Delta E(S_2-S_1) \sim 0.06$ eV, respectively. The regions where the electronic states are not well separated require non-adiabatic treatment. In contrast, the corresponding gap for the oligomer is $\Delta E(S_2-S_1) \sim 0.22$ eV, and the dynamics after crossing to the S_1 state can be denoted as essentially adiabatic.

3. a. Initial conditions and excitation. Again, the numerical results are presented without much explanation of the initial conditions (not until the methods section). From what I was able to understand, a high lying excited state is occupied by a laser pulse of 100fs duration. Which? The time axis in the figures is the time after the system has hopped to the "S2" state? Are all ensemble averages taking over this time variable, rather than the absolute time from excitation? This may be relevant, and should be clarified.

The system in the ground state configuration is instantaneously promoted to the excited state and non-adiabatic dynamics begins from this configuration for an entire ensemble of molecules spanning available conformational space. The initial dynamics on the excited state surface include the adiabatic vibrational relaxation toward the ES minimum before entering the NA region and NA transitions. The ensemble averages are taken over the same time variables unless otherwise noted by Δt . For example, the axis in Figure 3b $\Delta t_{S_2, S_1} = t - t_{\text{hop}}$ signifies the time from the S_2 to S_1 transitions where negative values are before the transition and positive values are after the transition, and 0 corresponds to the moment of transition. This is a convenient time variable that allows the trajectories to be synchronized because individual trajectories undergo transition at different times.

This discussion is reflected in the modified MS:

"...no superficial distinction between adiabatic and non-adiabatic regimes is made in our simulations, and the NEXMD trajectories are run for the entire duration of the dynamics starting from a ground state configuration instantaneously promoted to an excited state (Fig. S3). For some systems, the wave packet passes through the non-adiabatic region fast (e.g., oligomer) and the subsequent dynamics is essentially adiabatic. However, others (such as dendrimer or dimer) represent the case when the energy separation between excited states is comparable to the frequency of intramolecular motions and non-adiabatic dynamics persist for longer timescales."

We have clarified our time variables by making the following additions to the revised MS:

"Such delocalization first usually emerges at the moment of non-adiabatic transition ($\Delta t=0$)..."

And clarified in the caption to Figure 3:

"In order to synchronize the NA transitions among individual trajectories, we introduce the convenient time variable $\Delta t = t - t_{\text{hop}}$. The exact moment of NA transition, which varies among trajectories, is set to $\Delta t=0$. Negative values of Δt correspond to times before the NA transition and positive values represent times after the transition."

No actual values are given for the energy of the excitation; this should be presented, along with the

energy levels of the 10 excited states consider per systems. This is important for another reason, the subject of considerable debate in photosynthetic systems: does the width of the pulse create coherences between electronic states in the initial condition (i.e. is the bandwidth broad enough to excite superpositions of levels, including the two that are investigated in detail)? If so, how does this effect the results obtained by NEXMD?

In the revised manuscript, we have added the equilibrated absorption spectra and density of excited states (DOES) for each model system at room temperature (300K), new to this work Fig. S3 in the SI. The spectra show the contributions of the different excited states and the laser as a Gaussian centered at the excitation wavelength. Notably, in our simulations we are trying to illustrate our conceptual model as close as possible. The extensive results of our NEXMD simulations, in some cases under different excitation conditions are presented in our previous works Refs [39, 40, 41, 55] in revised MS.

Specific changes in the MS:

“However, others (such as dendrimer or dimer) represent the case when the energy separation between excited states is comparable to the frequency of intramolecular motions and non-adiabatic dynamics can persist for longer timescales. Consistent with the model in Fig. 1, the excitation is made to the low energy states that are less strongly overlapping compared to the dense energy manifold at higher energies that can be confirmed in the equilibrated absorption spectra in Fig. S3 showing the density of excited states along with the excitation energy for each system. The electronic character of the initial excitation is mostly uniform across the ensemble.”

Figure S3 Equilibrated absorption spectra and density of excited states (DOES) for each model system at room temperature (300K). The spectra show the contributions of individual excited

states, plotted as a distribution over the equilibrated ensemble, and the excitation wavelength. For each trajectory, only a single initial state is populated. Among the ensemble, the initial state may vary due to conformational disorder.

The second part of the Referee question is “does the width of the pulse create coherences between electronic states in the initial condition” is very interesting and it is not clear if the presented NEXMD simulation within the framework of the FSSH algorithm are able to fully address this aspect, particularly pertaining to experimental conditions.

We note that for each trajectory, only a single initial state is populated, not a superposition. The simulated pulse is Gaussian shaped with a width. For any excited state whose energy falls inside that pulse width, the oscillator strength is weighted by the value of the Gaussian (in this way states with large oscillator strength at the edge of the pulse may be weighted less and states with smaller oscillator strength in the center of the pulse may be weighted more). The weighted OS are normalized from 0 to 1 and a random number is generated so that the state whose weighted oscillator strength is largest will have the highest probability of being populated.

Among the ensemble, however, the initial state may vary from one trajectory to another due to the conformational disorder causing the excited state energies to shift within the pulse window. It is important to stress that excitation is not made to the high lying states where the energy manifold is dense. Instead, the excitation is made at low energy states that are less strongly overlapping to be consistent with the model in our Figure 1. In this way, the electronic character of the initial state is mostly uniform across the ensemble.

Because our trajectories are computed within the independent trajectory approximation, there is no interaction between individual trajectories. In this construction, the coherence between electronic states observed in the dynamics is necessarily at the single trajectory level. Multi-trajectory interactions are beyond the scope of the present NEXMD approach. We have been developing recently controlled approximations which goes beyond the Tully’s surface hopping approach toward interacting trajectories, namely, coherent Gaussian (White, A., Tretiak, S. & Mozyrsky, D. Coupled wave-packets for non-adiabatic molecular dynamics: a generalization of Gaussian wave-packet dynamics to multiple potential energy surfaces. *Chem. Sci.* 7, 4905 (2016); Makhov, D.V., Symonds, C., Fernandez-Alberti, S. & Shalashilin, D.V. Ab initio quantum direct dynamics simulations of ultrafast photochemistry with multiconfigurational Ehrenfest approach. *Chem. Phys.* 493, 200 (2017)). These tools are expected to provide a better answer to the Referee’s question on coherences created in experiment initially by a broad pulse.

We finally notice that while the dynamics in the ensemble introduces additional decoherence observed in the average, we do observe in-phase coherent vibronic dynamics across the entire ensemble of trajectories with slow decay for well over 100 femtoseconds at room temperature.

To address the Referee’s concern we modified the text:

“...as illustrated in the case of the nanohoop (see Fig. S7 in SI). An important spectroscopic observation is that the broad pulse may create coherences between electronic states in the

initial condition^{5,9-14}. These aspects invite further investigation by direct electronic dynamics modeling using advanced methodologies capable of describing interacting trajectories such as coherent Gaussian wave-packet approaches or multi-configurational methods^{60,61}.”

Presentation issues:

1. The manuscript begins with a definition of coherence which is rather vague. Off diagonal elements in a density matrix can depend (trivially) on a choice of basis, so the authors need to clarify precisely what they define as coherence (spatial, between eigenstates of some Hamiltonian...). For example, from the point that the "anti-symmetric" motion has been created on the lower PES and is out of the NA region, the dynamics of the system ("the sloshing") are essentially classical motions on an adiabatic surface, which for quantised nuclear degrees of freedom is a type of "coherent" motion, but one that is as close as possible to classical (this is the approximation used in the numerical method). "Coherence" of this kind, as might be generated by raman scattering onto a ground state wave packet can often persist on timescales $\gg 10$ -100fs at RT. A paper making a similar point to this (rapidly changing electronic character along a co-ordinate coupling two low lying states) is that of Tiwari 2013 (see below).

The BO approximation also means that no entanglement between the electronic and bath degrees of freedom exist, so care must be taken in the definition of "coherence". Many of the works involving quantum baths place considerable emphasis on the establishment of quantum correlations between the system and bath, which cannot be described in a semiclassical method.

The Reviewer raises an important question. Indeed, the word 'coherence' has been used in a variety of definitions across different research communities from quantum information processing to time-resolved spectroscopies to non-linear dynamics, etc. Subsequently, in the beginning of the introduction, we are not in a position to provide a rigorous definition of coherence that will encompass the extensive body of previous experimental and theoretical work to which we refer. Instead, later in the paper we define the coherence in the framework of the present study. The Referee is correct: in our case, we have the spatial coherence between the eigenstates of electronic molecular Hamiltonian, which are dynamically modulated by classical vibrational motions. However, we disagree that 'the "anti-symmetric" motion has been created on the lower PES .., the dynamics ... are essentially classical motions on an adiabatic surface'. Crossing the NA region is often not a singular event and the dynamics is strongly dependent on the system in question. Particularly, it is obvious in the case of multi-chromophore system in an 'intermediate' coupling regime where the NA dynamics persists for some time. An interesting question raised by the Referee is how coupling to classical degrees of freedom would translate to a "coherent" motion for quantized vibrations, which we cannot answer directly with our present computational capability. Instead, we refer to the previous body of work in the revised MS, including studies considering quantum baths.

To address these points, we have modified our article as follows:

“This phenomenon is ubiquitous as it follows from simple interplays between localizations and symmetries of the wavefunctions. Namely, non-adiabatic transitions between excited states induce the spatial coherence between the eigenstates of the electronic molecular Hamiltonian, which are dynamically modulated by classical vibrational motions. Since such

transitions are often not a singular event and can persist for some time, observed dynamics is strongly dependent on the system in question. We first present..."

"Thus, symmetries of the initial wavefunctions define the form of vibrational excitation emerging after electronic relaxation, which, in turn, controls wave-like localization-delocalization motion of the final wavefunction underpinning synchronous vibronic dynamics in the excited state. The dynamics of long-lived ground state wave-packets in photosynthetic light-harvesting antennas has already been reported in experiment¹⁹. To validate the proposed scenario in realistic materials, we further study four systems..."

2. The introduction begins by highlighting the "discovery" of coherent behaviour in photosynthetic systems (~2007) and later mentions that "oscillatory evolution" of wavefunctions can improve transport. They go on to discuss the change in thinking, from purely electronic coherences to a more complex interplay of electronic and vibrations processes. For the latter point of view, the authors point to [1,3], both from 2017, missing an enormous body of literature on this subject, particularly in photosynthesis, that made precisely this point (normally within models that explicitly treat the vibrations as quantum degrees of freedom). Some of the most highly cited works include the following (but there are many more excellent works on this subject, it is still an active subject)

We thank the Reviewer for bringing these relevant references to our attention and we have added them to the revised manuscript along with the following discussion:

"...The change in thinking towards more complex interaction between vibrations and electronic coherences was particularly prevalent in the realm of photobiology¹⁷, where commonly employed models treat vibrations as quantum degrees of freedom¹⁸⁻²⁵."

```
@article{christensson2012origin,  
title={Origin of long-lived coherences in light-harvesting complexes},  
author={Christensson, Niklas and Kauffmann, Harald F and Pullerits, Tonu and Mančal, Tomáš},  
journal={The Journal of Physical Chemistry B},  
volume={116},  
number={25},  
pages={7449--7454},  
year={2012},  
publisher={ACS Publications}  
}
```

```
@article{tiwari2013electronic,  
title={Electronic resonance with anticorrelated pigment vibrations drives photosynthetic energy transfer outside the adiabatic framework},  
author={Tiwari, Vivek and Peters, William K and Jonas, David M},  
journal={Proceedings of the National Academy of Sciences},  
volume={110},  
number={4},  
pages={1203--1208},  
year={2013},  
publisher={National Acad Sciences}
```

}

@article{kolli2012fundamental,
title={The fundamental role of quantized vibrations in coherent light harvesting by cryptophyte algae},
author={Kolli, Avinash and O'Reilly, Edward J and Scholes, Gregory D and Olaya-Castro, Alexandra},
journal={The Journal of chemical physics},
volume={137},
number={17},
pages={174109},
year={2012},
publisher={AIP}
}

@article{chin2013role,
title={The role of non-equilibrium vibrational structures in electronic coherence and recoherence in pigment--protein complexes},
author={Chin, AW and Prior, J and Rosenbach, R and Caycedo-Soler, F and Huelga, SF and Plenio, MB},
journal={Nature Physics},
volume={9},
number={2},
pages={113},
year={2013},
publisher={Nature Publishing Group}
}

@article{kreisbeck2012long,
title={Long-lived electronic coherence in dissipative exciton dynamics of light-harvesting complexes},
author={Kreisbeck, Christoph and Kramer, Tobias},
journal={The Journal of Physical Chemistry Letters},
volume={3},
number={19},
pages={2828--2833},
year={2012},
publisher={ACS Publications}
}

@article{falke2014coherent,
title={Coherent ultrafast charge transfer in an organic photovoltaic blend},
author={Falke, Sarah Maria and Rozzi, Carlo Andrea and Brida, Daniele and Maiuri, Margherita and Amato, Michele and Sommer, Ephraim and De Sio, Antonietta and Rubio, Angel and Cerullo, Giulio and Molinari, Elisa and others},
journal={Science},
volume={344},
number={6187},
pages={1001--1005},
year={2014},
publisher={American Association for the Advancement of Science}

}

```
@article{bakulin2016real,  
title={Real-time observation of multiexcitonic states in ultrafast singlet fission using coherent 2D electronic spectroscopy},  
author={Bakulin, Artem A and Morgan, Sarah E and Kehoe, Tom B and Wilson, Mark WB and Chin, Alex W and Zigmantas, Donatas and Egorova, Dussia and Rao, Akshay},  
journal={Nature chemistry},  
volume={8},  
number={1},  
pages={16},  
year={2016},  
publisher={Nature Publishing Group}  
}
```

```
@article{lim2015vibronic,  
title={Vibronic origin of long-lived coherence in an artificial molecular light harvester},  
author={Lim, James and Pale{\v{c}}ek, David and Caycedo-Soler, Felipe and Lincoln, Craig N and Prior, Javier and Von Berlepsch, Hans and Huelga, Susana F and Plenio, Martin B and Zigmantas, Donatas and Hauer, J{\u{u}}rgen},  
journal={Nature communications},  
volume={6},  
pages={ncomms8755},  
year={2015},  
publisher={Nature Publishing Group}  
}
```

3. The authors conclude that the insights arising from this work could be "exploited for design of functional organic materials". Could the authors please give one or two (brief, concrete) examples? Could the authors comment on how these results might translate to the case of nanoscale supramolecular complexes (most application of "coherence" are envisioned in systems containing multiple, not single, organic molecules)? Is there - generally - an obvious co-ordinate or adiabatic PES for a well-separated multi-chromophore system?

A general perception in the field is that molecular systems with strong electron-phonon coupling feature localized electronic states. Localized excitations can move by a hopping mechanism, that is not very efficient. Consequently, much work in the field was targeted to exploit possibilities of more efficient regime, where charge or energy carriers move, for example, via delocalized states toward band-transport regime. In fact, discussed experimental and theoretical studies aiming to clarify the role of coherence is an important sub-part of this more general effort.

Here we propose a general idea that an inevitable energy flow from electronic degrees of freedom to vibrations in the process of non-radiative relaxation and in the presence of strong electron-phonon coupling creates specific vibronic excitations that may propel the excited electronic state before localizing excitation. I.e., there exists a dynamical regime in which vibrations help to move the excitation from one place to another more efficiently. This mechanism may potentially provide an alternative view for and interpretation of existing and

future experiments. Notably, part of the molecular examples considered present cases of coupled multi-chromophore systems. However, across all examples studied, the dominance of such dynamical regime in terms of persistence and timescales is vastly different from system to system. Consequently, perhaps it is possible to direct synthetic efforts toward a desired function (such as specific directed funneling of excitons) by keeping in mind observed ultrafast dynamics of exciton-vibrations wavepacket. There is a significant body of work asking a question on existence of a specific 'coordinate or adiabatic PES for a multi-chromophore system' (for example, R-mode proposed by G. Zerbi). Instead, we are asking a question if there is a dynamical regime underpinning an efficient transport even in multi-chromophore systems with large disorder and strong electron-phonon coupling. How can we take advantage of this regime when going from regio-random to regio-regular form of polymers, in a specific packing of molecular crystals or specific pattern self-assembly of chromophores, or if Nature already uses it in frequently surprising arrangements of chromophores in light-harvesting membranes. These questions are up for an answer in future studies.

To account for the Referee suggestion and to reflect the above discussion we have added the following paragraphs to the text:

“Consequently, we conclude that these phenomena are omnipresent across a very broad range of molecular materials and may potentially provide an alternative interpretation of existing and future spectroscopic experiments. Namely, an inevitable energy flow from electronic degrees of freedom to vibrations in the process of non-radiative relaxation and in the presence of strong electron-phonon coupling creates specific vibrational excitations that spatially modulate the excited electronic state before localizing it into a 'self-trapped' excitation. Thus, there exists a dynamical regime in which vibrations may efficiently transfer the electronic excitation across molecular constituents. Across all examples studied, such dynamics are vastly different from system to system in terms of persistence and timescales including cases of coupled multi-chromophore systems. Consequently, it may be possible to achieve the desired function (such as specific directed funneling of excitons) by relying on observed ultrafast dynamics of exciton-vibrations (e.g., by seeking a dynamical regime underpinning an efficient transport in multi-chromophore systems with large disorder and strong electron-phonon coupling). Thus, these observed underlying physical principles can be further exploited for design of functional organic materials for various optoelectronic applications...”

In conclusion, I believe that the fundamental idea proposed in this paper is correct and of potentially high impact in this field, but revisions to this manuscript and a detailed response to my questions are required before I can recommend it for publication.

REVIEWERS' COMMENTS:

Reviewer #2 (Remarks to the Author):

The authors have given a very comprehensive and thoughtful response to my previous questions and remarks, and have generally convinced me that the mechanism that they discuss is a broad and novel-enough principle to be of wide interest. The new MS better highlights the underlying physics, and does so in a qualitative way that will be accessible to researchers from communities outside of PChem. I also think that they have made a good case in terms of distinguishing the present work from their previous studies, the key difference - for me - being the universality of the behaviour they find across examples.

It is a pleasure to recommend this improved MS for publication in Nature Communications.